# Assessing the Safety of Total Intravenous Anesthesia with Remimazolam in General Anesthesia for Transcatheter Aortic Valve Implantation of Severe Aortic Valve Stenosis: A Case Series

**DOI:** 10.3390/medicina58111680

**Published:** 2022-11-19

**Authors:** Yu-Yil Kim, Hyun-Joo Heo, Ji-Hye Lee, Hyung-Gu Cho, Geonbo Kim

**Affiliations:** Department of Anesthesiology and Pain Medicine, Presbyterian Medical Center, Jeonju 54987, Republic of Korea

**Keywords:** anesthetics, aortic stenosis, general anesthesia, intravenous, remimazolam, transcatheter aortic valve implantation

## Abstract

*Background and Objectives*: In patients with severe aortic stenosis (sAS), it is crucial to maintain hemodynamic stability during the induction and maintenance of general anesthesia for transcatheter aortic valve implantation (TAVI). In this study, we assessed the efficacy and safety of remimazolam in maintaining hemodynamic stability during anesthetic induction and maintenance. *Cases*: TAVI was performed on seven patients with sAS, and remimazolam was administered for total intravenous anesthesia (TIVA) of general anesthesia with induction (3.0 mg/kg/h) and maintenance (1.0 mg/kg/h). All patients underwent TAVI without major hemodynamic concerns and later recovered. *Conclusions*: Remimazolam can be safely used for induction and maintenance of general anesthesia in patients with sAS when performing TAVI.

## 1. Introduction

Aortic stenosis (AS) is one of the most common valvular heart diseases. In severe cases, medical management alone can lead to poor prognosis, and mechanical obstruction of the left ventricular outflow tract must be corrected with an aortic valve replacement (AVR) to improve survival. Transcatheter aortic valve implantation (TAVI) was first introduced in patients with inoperable or high surgical risk. Compared with surgical aortic valve replacement (SAVR), TAVI has a lower risk of major bleeding, acute kidney injury, and new-onset or worsening atrial fibrillation. Moreover, it is linked to a shorter length of hospital stay as well as lower in-hospital mortality and cost, and its application has been extended to patients with intermediate or even lower surgical risk [1,2].

TAVI is performed under general anesthesia (GA) and local anesthesia with or without conscious sedation (CS) [3]. Initially, TAVI has been performed under general anesthesia. As the operators’ experience increased and the valve systems was improved, the anesthetic modalities including local anesthesia and CS were adjusted to be relatively simplified. However, both modalities (GA and CS) may have advantages as well as disadvantages. Although local anesthesia with or without CS is currently preferred, optimal anesthesia for TAVI is still unclear. Consequently, patient factors, preference, and operator and TAVI center experience are important factors in determining the optimal anesthetic type [4].

The main concern of GA for TAVI is intraoperative hemodynamic management. It is important to maintain an appropriate cardiac output in patients with severe AS (sAS) during GA induction or maintenance by preventing potential hemodynamic instability, including hypotension. However, most intravenous anesthetic agents cause cardiovascular depression by dose-dependently reducing systemic vascular resistance (SVR) and cardiac contractility. Thus, intravenous anesthetic agents such as benzodiazepines (BZDs), ketamine, and propofol are commonly used for induction of GA, and volatile anesthetics or propofol are used for maintenance of GA in patients with sAS who underwent TAVI. Propofol, the most widely used intravenous anesthetic, should be used with extremely caution by titrating the dose for anesthesia induction and maintenance of anesthesia.

Remimazolam is an ultra-short-acting BZD and was developed based on the molecular structure of midazolam. It is a soft drug that is rapidly metabolized into inactive metabolites by nonspecific esterases in the body and can be used for anesthesia induction and maintenance [5]. Remimazolam is highly similar to midazolam in hemodynamic effects, and remimazolam is reported to have superior hemodynamic stability compared to other intravenous anesthetic agents, including propofol [6].

Previous reports have shown that remimazolam is used safely for anesthesia induction in patients with sAS [7,8]. However, there have been no reports that GA for TAVI was performed using TIVA with remimazolam. Thus, we report seven cases of TAVI that were stably performed with TIVA using remimazolam and remifentanil.

## 2. Case Presentations

We reviewed the patients’ medical records following ethics approval by our hospital Institutional Review Board (IRB, no. E2022-08), and the requirement for informed consent was waived by the IRB.

All patients underwent TAVI for symptomatic sAS except patient 3. Patient 3 was diagnosed with sAS in the preoperative evaluation for surgery of malignancy and underwent TAVI before surgery. The demographic and echocardiographic data of all patients are described in Table 1 and Table 2.

### Anesthesia

TAVI was performed under GA in all seven patients. TIVA using remimazolam and remifentanil was used for GA. No pre-medication was administrated, and the patients were monitored with a noninvasive blood pressure, 5-lead electrocardiogram, pulse oximeter at the operating room (OR). In addition, the patient state index (PSI, SedLine^®^, Masimo Corp, Irvine, CA, USA) was used for anesthetic depth monitoring. Before induction, preoxygenation was performed via a facial mask with 100% oxygen at 6 L/min for 2–3 min, and arterial cannulation was performed at the radial artery for invasive monitoring such as continuous arterial pressure, cardiac output, cardiac index, and stroke volume variation. Anesthesia was induced by continuous infusion (CI) of remimazolam, 1.0 mg/mL in sterile 0.9% sodium chloride, at 3.0 mg/kg/h. Remifentanil was infused at 1.5–3.0 ng/mL via target-controlled infusion (TCI) with plasma concentration using the Minto model. Once the patient’s modified observer’s assessment of alertness/sedation (MOAA/S) scale dropped to 1 or lower and the PSI dropped to 50 or lower, the remimazolam infusion was adjusted to 1.0 mg/kg/h, and rocuronium (0.6–1.0 mg/kg) was administrated intravenously. When the train-of-four count was 0 for two consecutive times, endotracheal intubation was performed using a video laryngoscope. Subsequently, external jugular vein cannulation was performed for fluid management and cardiogenic drug administration. A transesophageal echocardiography (TEE) device was inserted for early detection of procedural complications and assessment of the implanted valve function during TAVI. All patients had stable vital signs during induction. Anesthesia was maintained with remimazolam (0.6–1.5 mg/kg/h, CI) and remifentanil (0.5–2.0 ng/mL, TCI) with plasma concentration. Vital signs were stable throughout surgery, and the blood pressure and heart rate rapidly returned to normal levels following valve replacement using rapid ventricular pacing. To correct the hemodynamic instability of the patient during surgery, ephedrine, phenylephrine, and norepinephrine were prepared before the induction of anesthesia. In cases 1, 2, and 7, norepinephrine was used for hypotension that occurred when pretesting rapid ventricular pacing after placing the pacing wire; norepinephrine was maintained at a low dose (0.01–0.03 μg/kg/min) during balloon valvuloplasty and valve implantation to ensure hemodynamic stability. Norepinephrine was terminated when the vital signs were stabilized following valve replacement. In other cases, vasopressors and inotropes were not administered. All cases of TAVI were concluded without complications.

After the surgery, remimazolam and remifentanil infusions were discontinued, and neuromuscular block was reversed with sugammadex (2 mg/kg). Patients in cases 2–7 were provided with flumazenil (0.2–0.3 mg) after the surgery. Extubation was performed after the PSI increased to 80 or higher, and spontaneous breathing was recovered. The time from the end of surgery to the end of anesthesia was 10–20 min. The patients were transferred to the post-anesthesia care unit (PACU), where they showed stable vital signs. Furthermore, re-sedation in the PACU was not observed in patients who were used flumazenil. Patient 1 scored 3 on the MOAA/S but recovered immediately in the PACU. Upon discharge from the PACU, all patients had a post-anesthetic recovery score of 10. Patient 2 was transferred to the nephrology department for hemodialysis and chronic kidney disease (CKD) management due to postoperative exacerbation of CKD. Other patients were discharged on day 5 after TAVI. Patient 3 underwent a surgery for malignancy two weeks after TAVI. Table 3 shows the perioperative data of patients.

## 3. Discussion and Conclusions

TAVI was successfully performed without hemodynamic instability using TIVA with remimazolam in seven patients with sAS. No ephedrine and phenylephrine was used during induction and maintenance anesthesia, and low-dose norepinephrine alone was used for a short period during rapid ventricular pacing for valve implantation in three patients.

A number of studies, meta-analyses, and randomized clinical trials, comparing GA and local anesthesia with or without CS for TAVI, have been reported; however, the optimal anesthetic management is still considered controversial [4,9,10,11,12]. In most studies, the short and long-term mortalities of CS and GS for TAVI are similar, and CS is reported to have advantages in efficiency, such as shorter procedure time and shorter hospital stay. Local anesthesia with or without CS in TAVI is increasing, and some groups use CS exclusively. Some studies have reported that CS is used in 43.1–65.1% [13,14,15]. However, because GA is performed in approximately 50% of patients undergoing TAVI in clinical practice, research on methods and anesthetics of GA in TAVI should be continuously conducted.

Patient factors, preference (anesthesiologist, operator, and patient), and the TAVI team’s experience are important factors in determining the anesthetic management for TAVI. In this study, the inexperience of the operator and the TAVI center was the major reason for performing GA. The purpose of performing general anesthesia was as follows. First, it is to secure the facilitation and safety of the procedure by preventing the unexpected movement of the patient and maintaining constant respiration through mechanical ventilation. Second, it is to monitor the patient’s hemodynamic status, implanted valve function, and TAVI-related complications during the procedure using TEE.

GA for TAVI should be performed cautiously to prevent hemodynamic instability such as hypotension and tachycardia, which may lead to arrhythmias, myocardial ischemia and injury, heart failure, and death [16]. Most intravenous anesthetic agents induce dose-dependent cardiovascular depression by reducing systemic vascular resistance and cardiac contractility, which is exacerbated in patients with cardiac dysfunction and the elderly. Therefore, midazolam, ketamine, and propofol are commonly used as intravenous anesthetic agents.

In this study, remimazolam was used as an anesthetic. Similar to other intravenous anesthetic agents such as BZDs and propofol, remimazolam induces amnesia, sedation, hypnosis, unconsciousness, and some degree of immobility [5]. However, in contrast to midazolam, a well-known BZD, its context-sensitive decrement time is similar to that of propofol, and thus, it can be used for TIVA [6,17]. In addition, remimazolam has similar hemodynamic characteristics to midazolam, and has superior hemodynamic stability than propofol [6,18]. Chen et al. [19] reported that the incidence of hypotension in patients undergoing colonoscopy was lower among those sedated with a bolus injection of remimazolam (23.7%) than propofol (51.0%). Further, the continuous infusion of remimazolam in healthy male volunteers had moderate hemodynamic effects, decreasing the mean arterial blood pressure by 24 ± 6% (approximately 20 mmHg) and increasing the heart rate by 28 ± 15% (approximately 20 bpm); however, the clinical relevance of these effects was minimal [17]. Morita et al. [20] reported that among patients who underwent TIVA using remimazolam and propofol with remifentanil, the remimazolam group had fewer hemodynamic side effects. For the remimazolam group, anesthesia was induced using 6 mg/kg/h or 12 mg/kg/h and maintained with 1 mg/kg/h, and for the propofol group, anesthesia was induced using a slow bolus injection of propofol (2.0–2.5 mg/kg) and maintained with 4–10 mg/kg/h upon loss of consciousness. Remifentanil was used together in all groups. Blood pressure dropped in 22% and 49.3% of patients in the remimazolam and propofol groups, respectively. Such hemodynamic stability of remimazolam is expected to play a key role in preventing intraoperative hypotension. Some studies have reported hemodynamic stability of remimazolam in patients with severe AS or elderly patients [7,8,21]. These studies have reported that remimazolam is a safe and effective anesthetic for patients with sAS.

For induction of anesthesia, 6–12 mg/kg/h of remimazolam is recommended; however, anesthesia was slowly induced using 3 mg/kg/h of remimazolam to prevent unexpected hemodynamic changes in this study. Loss of consciousness was achieved successfully without using additional drugs in all patients, and hypotension or arrhythmias did not occur. Ephedrine, phenylephrine, and norepinephrine were prepared for prompt treatment to hemodynamic instability following anesthesia induction, maintenance, and procedure such as rapid ventricular pacing, balloon valvuloplasty, and valve implantation. During anesthesia, low-dose norepinephrine (0.01–0.03 μg/kg/min) was used in patients 1, 2, and 7 to treat hypotension that occurred during pretest after placing the pacing wire. Moreover, norepinephrine was discontinued after the valve implantation was completed and the patient’s vital signs stabilized. Other vasopressors or inotropes were not used. The cardiac index measured during anesthesia was relatively stable, and cardiac index improved in all patients after valve implantation.

Remimazolam has a similar context-sensitive decrement time to that of propofol. However, the time for recovery of consciousness after drug discontinuation is longer with remimazolam than propofol because of the shorter context-sensitive half time in plasma and longer context-sensitive half time in the effect site than propofol [6]. In contrast to propofol, however, the effects of remimazolam can be reversed with flumazenil, which can facilitate the recovery of consciousness. Flumazenil can be used to differentiate prolonged sedation and postoperative stroke related to TAVI. In patient 1, flumazenil was not used, and the patient had a MOAA/S score of 4–5 when transferred to the PACU. The MOAA/S score dropped to 3 after arriving at the PACU, and we considered the use of flumazenil or additional testing for stroke. However, the patient spontaneously recovered consciousness (MOAA/S of 5) after a few minutes. In patients 2–7, consciousness was recovered rapidly and completely in the OR using flumazenil, and no complications were observed. In all patients for which remimazolam was reversed with flumazenil, re-sedation was not observed in the PACU. However, the routine use of flumazenil for reverse of remimazolam should be cautioned as there are no studies on its adverse effects.

Anesthetic depth was monitored with the PSI based on EEG (GA range: 25–50) in this study. While research on EEG parameters for the anesthetic depth involving remimazolam is insufficient, several studies have reported that the bispectral index, patient state index, and narcotrend index are inappropriate for measuring the sedative effects of remimazolam [18,22,23]. In this study, the PSI was slightly high in patient 3 (PSI; 35–55) and patient 7 (PSI; 44–55); the PSI 55 was observed immediately after induction in both patients. However, the GA range of PSI was maintained by increasing remimazolam dose, and there were no cases of awareness or recall in any of the patients. Although PSI appears to reflect the anesthetic depth by remimazolam in this study, further research is required on remimazolam and anesthetic depth monitoring.

No postoperative complications such as remimazolam-related delirium or cognitive dysfunction were observed in this study. Patient 2 had a pre-existing stage 4 chronic kidney disease and was transferred to the nephrology department for hemodialysis and disease management due to an exacerbation of chronic kidney disease postoperatively. Other patients were discharged on postoperative day 5 without complications.

In all patients, remimazolam provided adequate anesthesia for induction and maintenance and maintained hemodynamic stability during surgery. Therefore, remimazolam as a TIVA can be safely used for induction and maintenance of GA in patients with sAS when performing TAVI. Further large-scale studies need to be conducted on this topic.

## Figures and Tables

**Table 1 medicina-58-01680-t001:** Demographic Data of patients.

Variables	Age, Years	Sex, M/F	Height/Weight, cm/kg	BMI, kg/m^2^	Comorbidities
Patient 1	84	F	142/56	27.8	Hypertension, tetraplegia due to traumatic spinal cord injury
Patient 2	69	M	155/52	21.6	Hypertension, atrial fibrillation, chronic kidney disease, cerebral infarction
Patient 3	76	M	163/64	24.2	Hypertension, type 2 diabetes
Patient 4	85	M	171/63	21.4	Hypertension, benign prostatic hyperplasia, chronic obstructive pulmonary disease
Patient 5	84	M	159/70	27.7	Hypertension, hyperlipidemia
Patient 6	67	M	165/67	24.6	Heart failure, type 2 diabetes
Patient 7	80	F	154/52	21.9	Hypertension, type 2 diabetes

BMI; body mass index.

**Table 2 medicina-58-01680-t002:** Echocardiographic Data of patients.

Variables	Ejection Fraction (%)	Max Velocity of Aortic Valve, m/s	Mean Pressure Gradient, mmHg	Aortic Valve Area, cm^2^
Patient 1	64	4.6	50	0.69
Patient 2	67	6.2	91	0.53
Patient 3	77	4.6	47	0.85
Patient 4	70	4.3	44	0.82
Patient 5	67	4.3	42	0.73
Patient 6	42	3.8	40	0.77
Patient 7	76	4.5	43	0.70

**Table 3 medicina-58-01680-t003:** Perioperative data of patients.

Variables	Patient 1	Patient 2	Patient 3	Patient 4	Patient 5	Patient 6	Patient 7
Remimazolam total dosage, mg	140	150	120	120	120	150	90
Remimazolam induction dose, mg/kg/h	3.0	3.0	3.0	3.0	3.0	3.0	3.0
Remimazolam maintenance dose, mg/kg/h	1.0	0.8–1.0	0.92–1.0	0.8–1.0	0.6–1.0	1.0–1.5	0.87–1.0
Anesthesia time, min	105	135	125	135	110	125	115
Operation time, min	85	90	85	105	70	95	90
Interval time between operation end and anesthesia end, min	10	15	20	10	15	10	15
Patient state index *	34–39	20–27	35–55	22–44	37–41	33–41	44–55
Cardiac index, L/min/m^2^	2.4–3.6	1.9–3.7	2.3–3.7	1.6–3.6	1.9–2.7	3.5–5.3	1.7–2.7
Use of cardiogenic drugs during anesthesia							
Ephedrine	No	No	No	No	No	No	No
phenylephrine	No	No	No	No	No	No	No
Norepinephrine	Yes	Yes	No	No	No	No	Yes
PACU time, min	45	70	55	100	75	140	85
PAR score, admission/discharge	9/10	10/10	10/10	10/10	10/10	10/10	10/10
Flumazenil, mg	No	0.2	0.3	0.3	0.3	0.3	0.3
Post-operative complication	No	Exacerbation of CKD	No	No	No	No	No

* This patient state index is the value from the induction of anesthesia to the discontinuation of the anesthetic agent. PACU: post-anesthetic care unit, PAR: post-anesthesia recovery, CKD: chronic kidney disease.

## Data Availability

The datasets of the current study are available from the corresponding author upon reasonable request.

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
