# Peer review of "Assessing the Safety of Total Intravenous Anesthesia with Remimazolam in General Anesthesia for Transcatheter Aortic Valve Implantation of Severe Aortic Valve Stenosis: A Case Series"

_medicina, 2022, doi:10.3390/medicina58111680_

Round 1
Reviewer 1 Report (Previous Reviewer 1)
This is a very well-written paper. The issue of sedation and anesthesia in TAVR patients is debatable. You wrote a very detailed paper on Remimazolam in general anesthesia for transcatheter aortic valve implantation. I believe the subject would interest anesthesiologists. The paper also has a good discussion regarding TAVR and anesthesia. It is interesting, detailed and precise.
Reviewer 2 Report (Previous Reviewer 2)
my concerns have been addressed
This manuscript is a resubmission of an earlier submission. The following is a list of the peer review reports and author responses from that submission.
Round 1
Reviewer 1 Report
Thank you for the opportunity to review this paper. It is very well written and interesting. However, the discussion is too long and a bit disorganized. I think this paper is worth publishing, but with a better discussion.
Author Response
We are grateful to you and the reviewers for your valuable time in reviewing our paper and providing valuable comments. The authors have carefully considered the comments and have done our best to address all comments. We hope the manuscript after careful revisions meet your high standards. The authors welcome any more constructive comments.
Below we provide the point-by-point responses. All revisions in the manuscript have been highlighted in red.
Thank you for your consideration. I look forward to hearing from you.
Sincerely,
Response to Reviewer 1
[General comment] Thank you for the opportunity to review this paper. It is very well written and interesting. However, the discussion is too long and a bit disorganized. I think this paper is worth publishing, but with a better discussion.
Response: We appreciate your comment. The discussion was written short and more organized.

Reviewer 2 Report
The present case report “Assessing the safety of total intravenous anesthesia with remimazolam in general anesthesia for transcatheter aortic valve im- 3 plantation of severe aortic valve stenosis: A case series” is presenting a cases series where the authors use remimazolam for MAC during TAVI in patients with severe aortic stenosis.
In general, the case series is targeting a very important topic. However, there are several points regarding to the conception of the study that need to be addressed and I have major concerns regarding the performed standards during the TAVI procedure.
Major:
Line 37f: “GA is generally induced using benzodiazepines (BZDs), etomidate, and opioids and maintained with a combination of inhalation anesthetics and opioids for patients with AS undergoing TAVI.” I absolutely disagree. There are various strategies for providing GA in cardiac patients and at our institution we don’t use Eto at all because of the side effects. Please change or use reverence for this statement. Maybe the SOLVE tavi studie should be cited.
Line 40: please explain why propofol needs to be used with caution.
Line 81 f: “the patients were connected 81 to a noninvasive blood pressure” please rephrase
Line 82: please explain what a PSI monitor is
Line 87: please specify what TCI calculation was used
What was the total Remimazolam dosage for introduction?
Line 91: please explain why invasive blood pressure monitoring was installed after induction of anesthesia. In my opinion induction of aneshtesia is a potential risk for patients suffering from severe AS.
Line 94: please explain why TOE was performed for TAVI
Author Response
We are grateful you and the reviewers for your valuable time in reviewing our paper and providing valuable comments. The authors have carefully considered the comments and have done our best to address all comments. We hope the manuscript after careful revisions meet your high standards. The authors welcome any more constructive comments.
Below we provide the point-by-point responses. All revisions in the manuscript have been highlighted in red.
Thank you for your consideration. I look forward to hearing from you.
Sincerely,
Response to Reviewer 2
[General comment] The present case report “Assessing the safety of total intravenous anesthesia with remimazolam in general anesthesia for transcatheter aortic valve implantation of severe aortic valve stenosis: A case series” is presenting a cases series where the authors use remimazolam for MAC during TAVI in patients with severe aortic stenosis.
In general, the case series is targeting a very important topic. However, there are several points regarding to the conception of the study that need to be addressed and I have major concerns regarding the performed standards during the TAVI procedure.
Response: We appreciate your comment. We have carefully considered the comments and tried our best to address every your comments.
[Comment 1] Line 37f: “GA is generally induced using benzodiazepines (BZDs), etomidate, and opioids and maintained with a combination of inhalation anesthetics and opioids for patients with AS undergoing TAVI.” I absolutely disagree. There are various strategies for providing GA in cardiac patients and at our institution we don’t use Eto at all because of the side effects. Please change or use reverence for this statement. Maybe the SOLVE tavi studie should be cited.
Response: We agree with the reviewer. Considering the hemodynamic stability of induction drugs, it was expressed that these drugs can be used. We revised the sentence as follows:
“Most intravenous anesthetic agents cause cardiovascular depression by dose-dependently reducing systemic vascular resistance (SVR) and cardiac contractility. Propofol, the most widely used intravenous anesthetic, also has these cardiovascular effects, thus, it must be used with extra caution by titrating the dose for anesthesia induction and maintenance.”
[Comment 2] Line 40: please explain why propofol needs to be used with caution.
Response: We also revised the sentence with considerate of comment 1 and 2. :
“Most intravenous anesthetic agents cause cardiovascular depression by dose-dependently reducing systemic vascular resistance (SVR) and cardiac contractility. Propofol, the most widely used intravenous anesthetic, also has these cardiovascular effects, thus, it must be used with extra caution by titrating the dose for anesthesia induction and maintenance.”
[Comment 3] Line 81 f: “the patients were connected to a noninvasive blood pressure” please rephrase
Response: We revised the sentence as follows:
“The patients were monitored noninvasive blood pressure, 5-lead electrocardiogram, pulse oximetry at the operating room (OR). “
[Comment 4] Line 82: please explain what a PSI monitor is
Response: We revised the sentence as follows:
“In addition, the patient state index (PSI, SedLine®, Masimo Corp, Irvine, CA, USA) was used for anesthetic depth monitoring.”
[Comment 5] Line 87: please specify what TCI calculation was used.
Response: We revised the sentence as follows:
“Remifentanil was infused at 1.5–3.0 ng/mL via target controlled infusion (TCI) with plasma concentration using the Minto model.”
[Comment 6] What was the total Remimazolam dosage for introduction?
Response: Unfortunately, we couldn’t measure the total dosage of remimazolam for induction
[Comment 7] Line 91: please explain why invasive blood pressure monitoring was installed after induction of anesthesia. In my opinion induction of aneshtesia is a potential risk for patients suffering from severe AS.
Response: That is an error when writing. Arterial cannulation was performed before anesthesia induction.
[Comment 8] Line 94: please explain why TOE was performed for TAVI
Response: We revised the sentence as follows:
“A transesophageal echocardiography (TEE) device was inserted for early detection of procedural complications and assessment of valvular functions during TAVI.”

Round 2
Reviewer 2 Report
Although I received a revised version there was no file where the authors addressed my concerns. However, I still don't understand why TAVI was performed under general anesthesia and not minimal anesthesia care and what TOE was used for.
Author Response
We are grateful the reviewers for your valuable time in reviewing our paper and providing valuable comments. The authors have carefully considered the comments and have done our best to address all comments. We hope the manuscript after careful revisions meet your high standards. The authors welcome any more constructive comments.
We seriously regret that the Round 1 authors' response notes file was not delivered to the reviewer, and we send it along with the Round 2 authors’ response notes file.
Below we provide responses. There are no changes to the manuscript in this revision.
Response to Reviewer 2
[Round 2]
[Reviewer comment] Although I received a revised version there was no file where the authors addressed my concerns. However, I still don't understand why TAVI was performed under general anesthesia and not minimal anesthesia care and what TOE was used for.
Response: First of all, we apologize that the Round 1 response file was not delivered, and we appreciate your Round 2 comment. We have carefully considered the comments and tried our best to address every your comments.
In the Round 1 revision, our explanation seems to have been insufficient. The reasons for performing general anesthesia and the purpose of using TEE are described as follows.
Initially, transcatheter aortic valve replacement or implantation (TAVR or TAVI) have been performed under general anesthesia. As the operator’s experience increased and the valve systems was improved, the anesthetic modalities including conscious sedation was changed to be relatively simple. However, both modalities (general anesthesia and conscious sedation) may have advantages as well as disadvantages. Consequently, patient factors, preference, and operator and TAVI center experience are important factors in determining the optimal anesthetic type. In this study, the operator was inexperienced, and therefore, general anesthesia was performed to prevent abrupt movement or respiratory depression of the patient, and to use transesophageal echocardiogram (TEE) during TAVI. Although TEE is not a routine procedure, it can be used for the purpose of early detecting procedural complications, evaluating paravalvular leakage, and reducing the amount of contrast agents. In this study, TEE was used more because the operator was inexperienced.
[reference; General versus local anesthesia with conscious sedation in transcatheter aortic valve implantation: the randomized SOLVE-TAVI trial. Circulation 2020; 142: 1437-1447]
